# Oregano Essential Oil versus Conventional Disinfectants against *Salmonella* Typhimurium and *Escherichia coli* O157:H7 Biofilms and Damage to Stainless-Steel Surfaces

**DOI:** 10.3390/pathogens12101245

**Published:** 2023-10-15

**Authors:** Jesus M. Luna-Solorza, J. Fernando Ayala-Zavala, M. Reynaldo Cruz-Valenzuela, Gustavo A. González-Aguilar, Ariadna T. Bernal-Mercado, M. Melissa Gutierrez-Pacheco, Brenda A. Silva-Espinoza

**Affiliations:** 1Centro de Investigación en Alimentación y Desarrollo, Asociación Civil, Carretera Gustavo Enrique Astiazarán Rosas, No. 46, Col. La Victoria, Hermosillo 83304, Sonora, Mexicojayala@ciad.mx (J.F.A.-Z.); reynaldo@ciad.mx (M.R.C.-V.); gustavo@ciad.mx (G.A.G.-A.); 2Departamento de Investigación y Posgrado en Alimentos, Universidad de Sonora. Blvd. Luis Encinas y Rosales S/N, Col. Centro, Hermosillo 83000, Sonora, Mexico; thalia.bernal@unison.mx; 3Ciencias de la Salud, Universidad Estatal de Sonora, Campus San Luis Rio Colorado, Carretera San Luis Rio Colorado-Sonoyta Km 6.5. Col. Industrial CP, San Luis Río Colorado 83430, Sonora, Mexico; melissa.gtzpacheco@gmail.mx

**Keywords:** essential oil, stainless-steel damage, bacterial adhesion, *E. coli*, *S.* Typhimurium, biofilm, oregano oil

## Abstract

This study compared the effect of oregano essential oil versus sodium hypochlorite, hydrogen peroxide, and benzalkonium chloride against the viability of adhered *Salmonella* Typhimurium and *Escherichia coli* O157:H7 on 304 stainless steel. Oregano essential oil was effective in disrupting the biofilms of both bacteria at concentrations ranging from 0.15 to 0.52 mg mL^−1^. In addition, damage to stainless-steel surfaces following disinfection treatments was assessed by weight loss analysis and via visual inspection using light microscopy. Compared to the other treatments, oregano oil caused the least damage to stainless steel (~0.001% weight loss), whereas sodium hypochlorite caused the most severe damage (0.00817% weight loss) when applied at 0.5 mg mL^−1^. Moreover, oregano oil also had an apparent protective impact on the stainless steel as weight losses were less than for the control surfaces (distilled water only). On the other hand, sodium hypochlorite caused the most severe damage to stainless steel (0.00817% weight loss). In conclusion, oregano oil eliminated monoculture biofilms of two important foodborne pathogens on 304 stainless-steel surfaces, while at the same time minimizing damage to the surfaces compared with conventional disinfectant treatments.

## 1. Introduction

Bacterial disinfection of stainless-steel surfaces in the food industry represents a challenge, considering the damage attributed to this process. In addition, a reduced efficacy of disinfectants has been related to biofilms formed by adhered bacteria that live embedded in an extracellular polymeric matrix [1]. Compared to their planktonic counterparts, the resistance of biofilm-associated bacteria to a particular disinfectant can be a thousand times greater and, consequently, enhance the risk of foodborne illness [2]. It is believed that biofilms have led to more than four thousand cases of foodborne illness in the last five years, highlighting *Salmonella* Typhimurium and *Escherichia coli* as the leading species [3]. *S.* Typhimurium and *E. coli* O157:H7 are harmful foodborne pathogens causing salmonellosis and severe food poisoning, with symptoms like diarrhea, fever, and vomiting. *E. coli* O157:H7 produces a potent toxin leading to life-threatening conditions, especially in vulnerable individuals. These pathogens contaminate various foods, posing serious health risks if safety measures are not followed and resulting in severe illnesses, hospitalizations, and fatalities. This highlights the urgent need for strict food safety protocols and public health awareness. It has also been reported that some industries use sanitizers in excess, damaging surfaces and increasing costs [4,5]. The inappropriate use of disinfectants can also be found in the food industry [6,7], where many factors impact the development of biofilms, including moisture levels, available nutrients, and temperature fluctuations [8]. Since stainless steel is the preferred material for the fabrication of food processing equipment, it is important to seek effective and friendly sanitizing agents to control microbial contamination while at the same time minimizing damage to these surfaces.

Previous studies in this area demonstrated the antibiofilm efficacy of various synthetic and natural compounds [9,10,11], while other studies evaluated the effect of disinfectants on the integrity of surfaces [5,12,13]. One natural strategy is the use of enzymes; however, they do not affect bacterial viability, while synthetic disinfectants are rejected by consumers due to increasing awareness of their potential health and environmental implications. These studies also showed that chloride and acidic pH promote corrosion. In contrast, they evidenced that the essential oil of *Artemisia vulgaris* had an anticorrosive effect in oxidizing conditions; however, they did not measure the antibiofilm effect [14]. On the other hand, Bouyanzer and Hammouti [15] showed that *Artemisia* oil acted as a corrosion inhibitor of stainless-steel and they proposed that the oil adsorbs on the metal, blocks reaction sites and protects the surface from the acidic medium. However, these effects were not evaluated in the presence of microorganisms or during the simulation of an operating sanitation procedure, which would generate a more realistic picture of the complexity of food environments. Additionally, other types of essential oils could also have these benefits.

Essential oils derived from plants are gaining attention in food safety due to their natural antimicrobial properties. These oils disrupt bacterial biofilms on surfaces by interfering with bacterial adherence and communication [16]. Oregano (*Lippia graveolens*) essential oil (OEO) has a similar composition to *Artemisia*; both oils share camphene, *p*-cymene, α-pinene, α-thujene, and other terpenes [14,17]. OEO showed a bactericidal effect against Gram-negative and Gram-positive bacteria at 0.025–4 mg mL^−1^ [18]. The same study explained that the antimicrobial nature of OEO is due to the hydrophobic nature of terpenes such as carvacrol and thymol, which facilitate its interaction with bacterial membranes, causing the loss of functionality and viability. On the other hand, no studies evaluated OEO as an anticorrosive agent; however, this capacity could be expected due to its composition. Therefore, the objective of the work presented here was to compare the effect of OEO with sodium hypochlorite (NaClO), benzalkonium chloride (Benz), and hydrogen peroxide (H_2_O_2_) on biofilms of *S.* Typhimurium and *E. coli* O157:H7 and its corrosive damage to stainless steel.

## 2. Materials and Methods

### 2.1. Antibacterial Compounds

OEO (*Lippia graveolens*) was purchased from “Ore aceite de oregano” (Saucillo, Chihuahua, Mexico). Our research team had previously examined the chemical composition of the OEO [19]. It is important to note that OEO was predominantly composed of carvacrol (47.4%) and thymol (3%). Sodium hypochlorite (NaClO), hydrogen peroxide (H_2_O_2_), and benzalkonium chloride (Benz) were acquired from local stores in Hermosillo, Sonora, Mexico.

### 2.2. Minimum Inhibitory (MIC) and Bactericidal Concentrations (MBC) of OEO, NaClO, H_2_O_2_, and Benz against Planktonic S. Typhimurium and E. coli O157:H7

Bacteria were revived in Luria–Bertani broth (LB) at 37 °C for 18 h; then, the inoculum of each strain was adjusted by reference to an OD of 0.1 at 600 nm using a microplate reader Fluostar Omega (BMG Labtech, Chicago, IL, USA) and further diluted with LB broth to achieve approximately 1 × 10^6^ CFU mL^−1^. The MIC and MBC for each tested compound were estimated using the broth microdilution method by applying a concentration range of 0–1 mg mL^−1^ for each disinfectant. Serial dilutions of the treatments were prepared in LB broth, and 295 μL of these were mixed with 5 μL of the adjusted inoculum in Costar 96-well microtiter plates, and then incubated at 37 °C for 24 h. The MIC was determined as the lowest concentration of each compound to inhibit visible growth of the inocula. On the other hand, 10 μL of the MIC and two higher concentrations were inoculated on LB agar and incubated at 37 °C for 24 h. MBC was defined as the lowest concentration in which no colonies appeared on the LB agar medium [20]. These experiments were repeated in triplicate.

### 2.3. Cellular Adhesion of S. Typhimurium and E. coli O157:H7 on Stainless Steel during Incubation Time

The cellular adhesion of *Salmonella* Typhimurium ATCC 14028 and *Escherichia coli* O157:H7 ATCC 43890 on 304 stainless-steel (2.4 cm^2^) was obtained following the method described by Jadhav, et al. [21] with some modifications. Under sterile conditions, the stainless-steel coupons were placed in test tubes with LB broth, then inoculated with 1 × 10^6^ CFU mL^−1^ of each bacterium. The inoculated test tubes with the metal coupons were incubated at 37 °C, and adhered viable cells were counted (Log CFU cm^−2^) at 15 min and then at 2, 4, 6, 8, 24, 48, 72, and 144 h. This quantification was achieved by immersing the coupons in a 0.9% sodium chloride solution and applying an ultrasonic bath (42 kHz, Branson 2510 Ultrasonic Danbury, CT, USA) for 5 min. Serial dilutions were made to this suspension before plating on Bismuth sulfite agar for *S.* Typhimurium and MacConkey agar with sorbitol for *E. coli* O157:H7. This experiment was repeated three times.

### 2.4. Minimum Biofilm Inhibitory (MBIC) and Minimum Biofilm Eradication Concentrations (MBEC) of OEO, NaClO, H_2_O_2_, and Benz against S. Typhimurium and E. coli O157:H7

A series of test tubes, each containing 5 mL of LB broth with different concentrations of disinfectant, were prepared for each of the test compounds. To determine the MBIC values, a stainless-steel coupon was aseptically added to each tube, which were then inoculated with appropriate bacterial culture (1 × 10^6^ CFU mL^−1^) and subsequently incubated at 37 °C for 2 h. The method used to count the adhered viable cells in the biofilms was as described in the previous section. The concentration that completely inhibited the bacteria’s adhesion to the stainless steel was considered the MBIC [22].

The technique recommended by Chamdit and Siripermpool [23] was used with some modifications to determine the MBEC. Biofilms of *S.* Typhimurium and *E. coli* O157:H7 were preformed on stainless-steel coupons after 24 h of incubation at 37 °C. Preformed biofilms were washed with 0.9% sodium chloride solution (pH 6) to remove the weakly adhered cells and then exposed for 1 h to different concentrations (0 to 4 mg mL^−1^) of antibacterial compounds. Viable adhered cells were enumerated as described above. The MBEC was considered the concentration that eliminated 100% of the viable cells attached to the stainless-steel.

### 2.5. Swimming and Swarming Motility of Treated S. Typhimurium and E. coli O157:H7

The effect of antimicrobial compounds on the swimming and swarming motility of *S.* Typhimurium and *E. coli* O157:H7 was evaluated using the soft agar method with 0.3% and 0.5% agar (for swimming and swarming, respectively). Bismuth sulfite agar for *S.* Typhimurium and MacConkey agar for *E. coli* O157:H7 were added with the MBIC obtained previously. This agar was inoculated (1 × 10^6^ CFU mL^−1^) in the center and incubated at 37 °C for 24 h. After the incubation, the motility halo was measured, and results were expressed as % [24]. Bacteria viability (Log CFU mL^−1^) was evaluated to discard that the treatment effect on motility was due to its affection.

### 2.6. Weight Loss and Apparent Damage to Stainless Steel in Contact with Antibacterial Compounds

Two different experiments were carried out to prove the effectiveness of disinfectants on stainless steel. Coupons were immersed in the treatments and antibacterial agents were applied with brushing. The latter simulated a sanitation procedure, which consisted of using the antimicrobial compounds (0 to 1.2 mg mL^−1^) for 20 min, followed by a one-minute brushing and finally a water rinse. This process was repeated 90 times, which represents 30 days of the cleaning process; it was also applied to remove previously formed biofilms. The other experiment entailed exposing the coupons to different antimicrobial compounds (0 to 1.2 mg mL^−1^) for six weeks [12]. The weight loss (WL) of stainless steel was measured with a Sartorius MSE2255100-DU analytical balance, Goettingen, Germany, whereby coupons were weighed before and after treatments using the following formula: WL (%) = [(initial weight − final weight)/initial weight] × 100. In addition, the apparent damage to stainless steel exposed to sodium hypochlorite (0.4 to 1.2 mg mL^−1^) for six weeks was visualized with a RoHS digital microscope taking microphotographs before and after six weeks.

### 2.7. Experimental Designs and Statistical Analysis

A complete randomized design was used in all experiments. For cellular adhesion on stainless steel at different times, the factor was incubation time (h) and the response was the number of viable bacteria adhered to coupons (Log CFU cm^−2^). In the motility test, the factor was antimicrobial compounds and the response variable was the percentage of motility and cellular viability. For both experiments, an analysis of variance (ANOVA) was performed, followed by a Tukey–Kramer test (*p* < 0.05). On the other hand, a t-analysis was used to compare every treatment with a control in the metal weight loss response. An analysis of variance (ANOVA) was performed to evaluate the effect of the sanitation procedure and a multiple comparison test was conducted with the Tukey–Kramer test (*p* < 0.05), in which the factors were disinfectant concentrations applied with brushing and the unbrushed control, and the response was the number of adhered cells (Log CFU cm^−2^). All statistical analyses were performed using NSCC statistical software version 2012.

## 3. Results

### 3.1. Adhesion of S. Typhimurium and E. coli O157:H7 on Stainless-Steel

Figure 1 shows the number of *S.* Typhimurium and *E. coli* O157:H7 adhered to stainless-steel coupons at different times over the course of the incubation period at 37 °C. It can be observed that *S.* Typhimurium began its adhesion after 15 min (3.93 Log CFU cm^−2^), while *E. coli* adhered after two hours (2 Log CFU cm^−2^). An exponential increase in the number of adhered cells was observed for *S.* Typhimurium during the first 24 h of incubation, where significant differences (*p* < 0.05) were found between 2 h (4.39 CFU cm^−2^), 8 h (5.11 Log CFU cm^−2^), 24 h (5.5 Log CFU cm^−2^), and 48 h (5.54 Log CFU cm^−2^). In contrast, an exponential increase in *E. coli* adhesion occurred from 2 h to 6 h, with significant differences (*p* < 0.05) between 2 h (2 Log CFU cm^−2^), 4 h (3.19 Log CFU cm^−2^), and 6 h (4.11 Log CFU cm^−2^). These results evidenced the short time required for the cells to adhere to the stainless steel. Because both bacteria formed mature biofilms after 24 h, this time was selected to determine the minimum biofilm eradication concentration (MBEC).

### 3.2. OEO, NaClO, H_2_O_2_, and Benz against Planktonic S. Typhimurium and E. coli O157:H7

The antibacterial capacity of each compound is shown in Table 1. All antimicrobial compounds tested here inhibited and/or eradicated the planktonic cells and biofilms of *S.* Typhimurium and *E. coli* O157:H7. Benz was the most effective treatment (0.016 to 0.05 mg mL^−1^), and NaClO was the least effective (0.32 to 1.2 mg mL^−1^). H_2_O_2_ was the second most effective (0.03 to 0.36 mg mL^−1^), followed by OEO (0.15 to 0.6 mg mL^−1^). *S*. Typhimurium was more resistant to NaClO and less resistant to H_2_O_2_ than *E. coli* O157:H7. At the same time, both bacteria were equally susceptible to Benz. In general, a lower concentration of all treatments was required to inhibit biofilm formation compared to the eradication of preformed biofilms and the viability of planktonic cells of both bacteria. It is important to highlight that MIC denotes the lowest concentration of an antimicrobial agent that hampers bacterial growth without eradicating all bacteria. Conversely, MBC refers to the minimum concentration necessary to completely exterminate every bacterium in a population, leaving no survivors behind. These results also showed that a higher concentration was necessary to eradicate biofilms (MBEC) than planktonic state bacteria (MBC).

### 3.3. Swimming and Swarming Motility of S. Typhimurium and E. coli O157:H7

The swimming and swarming motility, along with cell viability, of *S.* Typhimurium and *E. coli* O157:H7 exposed to antibacterial agents are shown in Figure 2. All compounds inhibited 100% of *swimming* and *swarming* motility compared to untreated bacteria. The MBIC of OEO did not affect the viability of *S.* Typhimurium (*p* > 0.05), while MBICs of NaClO and H_2_O_2_ had a significant effect (*p* < 0.05) on cell viability. However, this reduction did not change this bacterium’s growth since it showed higher counts than the initially added inoculum. A similar observation was made in the case of *E. coli* O157:H7; H_2_O_2_ did not decrease cell viability during the experiment (*p* > 0.05), while OEO and NaClO were found to significantly reduce viability (*p* < 0.05); however, this reduction was not lower than the initial inoculum level. On the other hand, Benz significantly decreased the viability of both bacteria tested here, thus preventing their proliferation, as evidenced by their lower counts relative to the initial inoculum.

### 3.4. Effect of Antibacterial Agents Combined with Brushing on Viable Adhered Bacteria, Weight Loss, and Apparent Damage to Stainless Steel

Figure 3 shows the number of adhered viable cells of *S.* Typhimurium and *E. coli* O157:H7 after the prescribed sanitation procedure. All four compounds were found to completely eliminate the preformed biofilm at the concentrations tested. Moreover, compared to control without brushing, biofilm removal was also affected by brushing by itself (*p* < 0.05). However, this mechanical treatment was insufficient to remove substantial amounts of the preformed biofilms of *S*. Typhimurium and *E. coli* O157:H7, as reductions of only 0.8 Log CFU cm^−2^ and 0.3 Log CFU cm^−2^ were observed in the two pathogens, respectively.

Figure 4 shows the weight loss of stainless-steel coupons exposed to different concentrations of OEO, Benz, NaClO, and H_2_O_2_. NaClO was the treatment that most significantly affected weight loss; a concentration of 0.5 mg mL^−1^ caused the highest weight loss (0.00817%). NaClO at 1.2, 0.40, and 0.45 mg mL^−1^ caused weight loss values of 0.0028%, 0.00205%, and 0.00175%, respectively. Moreover, the weight loss caused by NaClO at 1 mg mL^−1^ (0.00351%) was the second-highest loss; it was statistically significant (*p* > 0.05). These effects on weight loss were visualized by microscopy, where the apparent damage to stainless-steel surfaces caused by NaClO treatments was readily observed (Figure 5). In Figure 5, the difference in the surface of stainless steel before and after exposure to various concentrations of NaClO for six weeks can be observed. Stains and scratches (indicated by arrows) which were not present before exposing the stainless steel to NaClO are visible in the figure. Therefore, it is reasonable to conclude that this damage was caused by the presence of NaClO. In agreement with the weight loss data, NaClO at 0.5 mg mL^−1^ appeared to cause the most severe observable damage to the stainless steel, followed by 0.4 and 1 mg mL^−1^, while concentrations of 1.2 and 0.45 mg mL^−1^ had the same apparent damage.

In the second experiment assessing damage caused by disinfectant treatments, the weight loss of stainless-steel coupons after 90 cleaning cycles and various concentrations of OEO, Benz, NaClO, and H_2_O_2_ was compared, as shown in Figure 6. NaClO at 1.2, 0.5, and 0.32 mg mL^−1^ caused weight losses of 0.0024%, 0.0018%, and 0.003%, respectively, while 0.05 and 0.02 mg mL^−1^ of Benz caused losses of 0.0018 and 0.002% and H_2_O_2_ at 0.36 mg mL^−1^ caused a 0.0025% loss in weight with respect to the control (*p* < 0.05). On the other hand, weight losses for OEO at 0.6, 0.26, and 0.15 mg mL^−1^ were 0.00008, 0.00003, and 0.00012%, while Benz at 0.16 mg mL^−1^ caused 0.0018%, and H_2_O_2_ at 0.16, and 0.03 mg mL^−1^ caused 0.0019 and 0.0024% weight loss, which was similar to the control (*p* > 0.05).

## 4. Discussion

Our results showing that the tested disinfectants effectively inhibited and eliminated planktonic and biofilm bacteria and are in agreement with previous studies [25,26,27,28,29,30]. Indeed, the concentrations obtained in our results are similar to those in other studies that used essential oils against the same bacteria [31]. However, there are no studies that evaluated their effect on surface damage. Benz needed concentrations up to 80% lower to achieve inhibition and elimination of *S.* Typhimurium [32]. Benz is known for interacting with cells’ negative charges, destabilizing them and causing viability loss, while H_2_O_2_ and NaClO can oxidize the bacterial membrane and cell wall, also causing viability loss. Finally, the antibacterial capacity of carvacrol and thymol, the main components of OEO, are known to disrupt the outer membranes of Gram-negative bacteria, thereby releasing lipopolysaccharides and increasing permeability to ATP [33]. The compounds in OEO have been previously demonstrated to impact biofilms formed on stainless steel by bacteria, such as Pseudomonas aeruginosa. This effect is attributed to their disruption of bacterial quorum sensing, leading to an interruption in the initial phase of biofilm formation [34]. The low effectiveness of NaClO against these bacteria is relevant since this disinfectant is one of the most used in the food industry.

All compounds affected bacterial motility without affecting viability, making these disinfectants good alternatives for disrupting this virulence factor. Because motility guides *S.* Typhimurium and *E. coli* O157:H7 to adhere to different surfaces, this effect can be considered a part of the disinfectants’ mode of action against biofilm formation. In addition, brushing improved the efficacy of the antimicrobial compounds in terms of reducing the number of viable adhered bacteria on the stainless-steel surface. Immersion treatments required 60 min to eradicate the preformed biofilms, while brushing required 20 min. This combination could be attributed to a synergistic effect with brushing that weakened the biofilm, allowing the antibacterial agent to reach embedded bacteria. In our search of the literature, no studies were found on the action of brushing in conjunction with these types of antimicrobial compounds against biofilms of foodborne pathogens. The fact that the sanitation procedures were effective led to evaluating their effect on the stainless-steel surfaces.

The weight losses exhibited by the stainless steel in these experiments were not dose-dependent. Different weight loss percentages were found between the experiment involving coupons treated by immersion in antibacterial agents for 6 weeks and that involving coupons treated with 90 cleaning cycles. NaClO was not the only treatment that caused weight loss in the stainless-steel (0.22 to 0.30 mg), as both Benz and H_2_O_2_ also managed to affect the metal, resulting in significant (*p* < 0.05) reductions in its weight (0.21 to 0.28 mg). Moreover, the results obtained from these trials also demonstrated that brushing combined with antibacterial agents negatively impacted stainless-steel structure, causing damage.

Waters, Tatum, and Hung [12] found that chlorine at 0.040 mg mL^−1^ caused weight loss values of 0.1 to 2 mg to 316 stainless-steel after 7 weeks of exposure. It is noteworthy that 316 stainless-steel contains molybdenum, which makes it more resistant to damage than 304 grade. In the present study, NaClO at 1.2, 1.0, 0.6, 0.5, 0.45, 0.4 and 0.32 mg mL^−1^ was found to cause weight losses of 0.32, 0.41, 0.19, 0.94, 0.20, 0.23, and 0.16 mg to 304 stainless steel, respectively; however, these were similar to values previously reported in 316 grade stainless-steel [12]. On the other hand, Ayebah and Hung [4] found that 304 stainless steel treated with calcium hypochlorite (2.82 mg mL^−1^) lost 1 mg of weight daily over an 8-day period, thus showing higher values than the present study. 

In the present study, NaClO caused the highest weight loss values and apparent visible damage to stainless steel. The same negative effect to stainless-steel surfaces, but to a lesser degree, was also observed for both Benz and H_2_O_2_, which could represent an extra expense for facilities using such sanitizers in their cleaning operations. Moreover, since NaClO was also found to be a less effective antibacterial agent against biofilms, this suggests a real need for the consideration of alternative disinfectants for such surfaces. In contrast, OEO was found to effectively control biofilm development by inhibiting the adherence of bacterial cells as well as disinfecting surfaces harboring preformed biofilms, without deteriorating the stainless-steel surfaces. Furthermore, OEO also displayed a protective capacity for stainless steel as it resulted in lower weight losses (0.005 to 0.03 mg) and apparent damage than control surfaces treated only with distilled water. This result may in part be due to the hydrophobic nature of stainless-steel surfaces themselves, which allows for a more specific interaction between the oil and metal.

## 5. Conclusions

This study probed the antibacterial efficacies and collateral material impacts of various disinfecting agents, namely sodium hypochlorite (NaClO), oregano essential oil (OEO), and others, against *E. coli* O157:H7 and *S.* Typhimurium biofilms on stainless-steel surfaces. Notably, NaClO necessitated elevated concentrations for effective bacterial eradication but concurrently imposed detrimental effects on stainless steel. Conversely, OEO demonstrated a notable antibacterial efficacy at minimized dosages, while concurrently exhibiting no deleterious impact on the stainless-steel substrates. Additionally, while brushing enhanced antibacterial efficacy, it proved insufficient as a standalone disinfection strategy. The findings underscore OEO’s potential as a pivotal disinfectant in food industry contexts, warranting further investigation into its mechanistic antibacterial and anticorrosive properties to elucidate potential applicabilities in holistic, efficacious microbial and material management within the sector.

## Figures and Tables

**Figure 1 pathogens-12-01245-f001:**
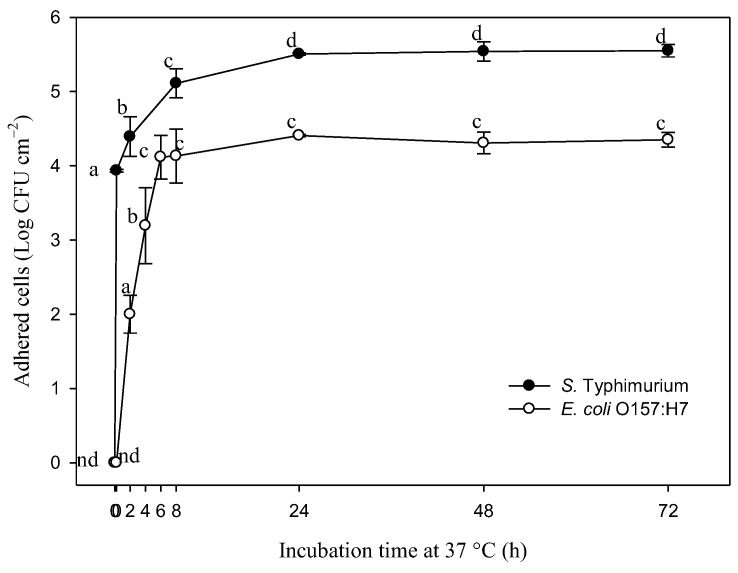
Changes in adhered viable cells of *S.* Typhimurium and *E. coli* O157:H7 on stainless-steel coupons during incubation time at 37 °C. Different letters indicate significant differences among times per bacterium (*p* < 0.05). nd: no adhesion detected.

**Figure 2 pathogens-12-01245-f002:**
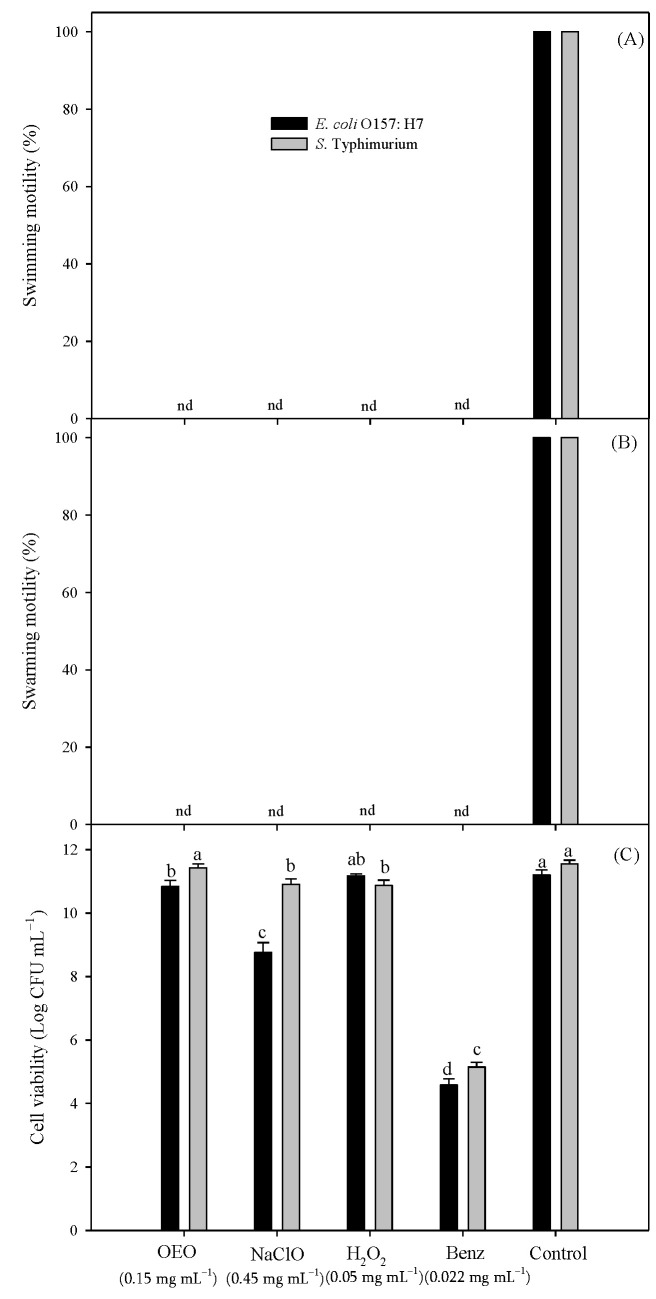
Effect of OEO, NaClO, Benz, and H_2_O_2_ against swimming (**A**), swarming (**B**), and cell viability (**C**) of *E. coli* O157:H7 and *S*. Typhimurium, respectively. Different letters mean significant differences among treatments per bacterium (*p* < 0.05). nd: no motility detected.

**Figure 3 pathogens-12-01245-f003:**
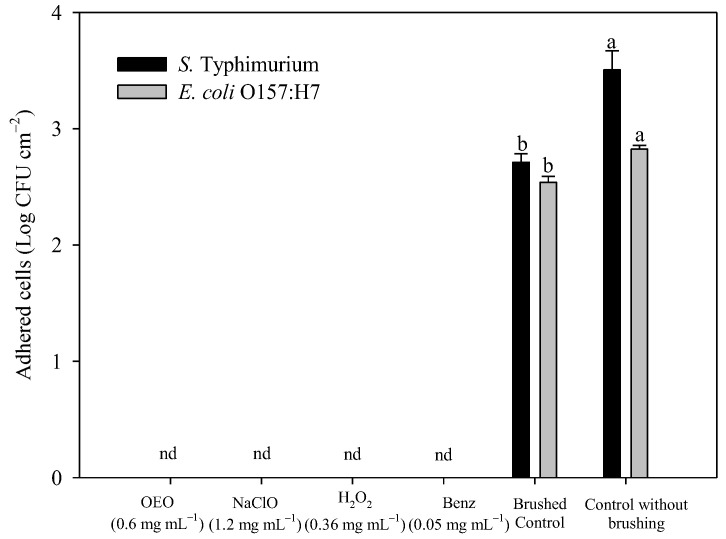
*S.* Typhimurium and *E. coli* O157:H7 adhered on stainless steel after sanitation with OEO, NaClO, H_2_O_2_, and Benz. nd: non detected. Different letters mean significant differences among treatments per bacterium (*p* < 0.05).

**Figure 4 pathogens-12-01245-f004:**
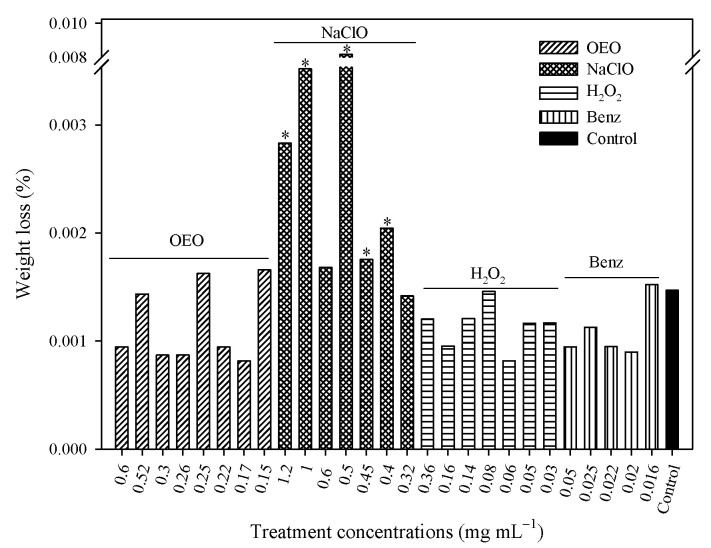
Weight loss of 304 stainless-steel exposed to different concentrations of OEO, NaClO, H_2_O_2_ and Benz for six weeks. *: Significant differences with the control (*p* < 0.05).

**Figure 5 pathogens-12-01245-f005:**
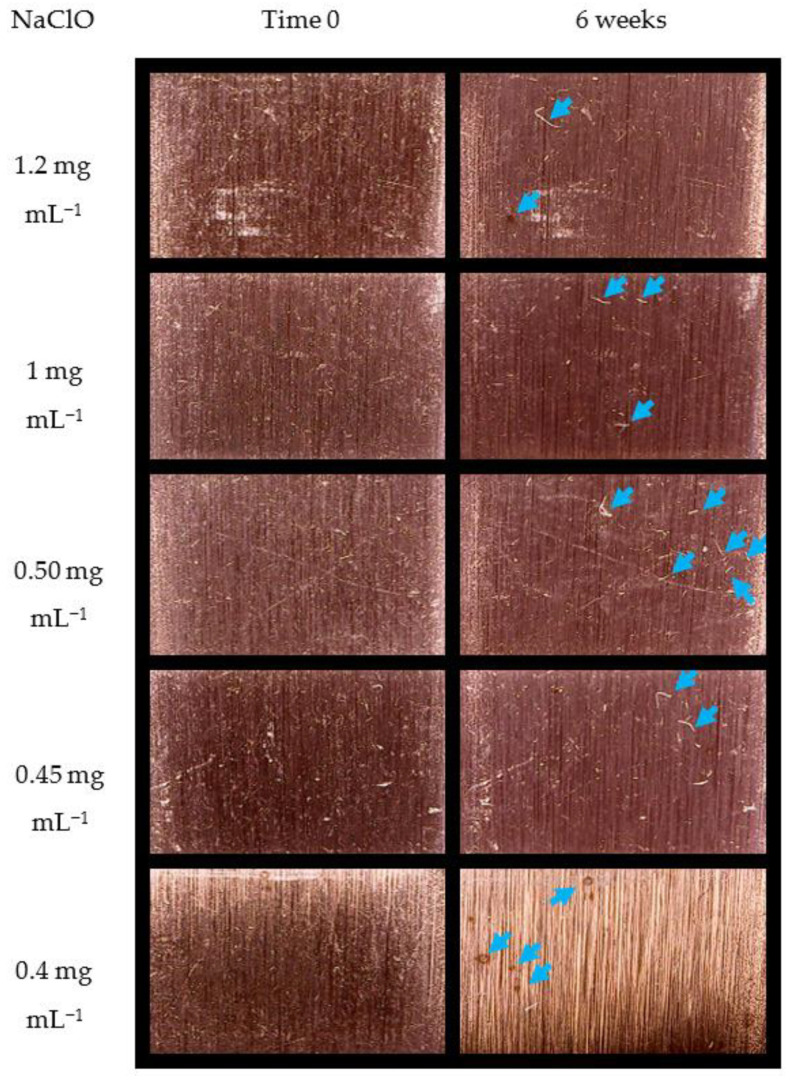
Photographs of 304 stainless-steel coupons at 0 and 6 weeks of exposure to NaClO.

**Figure 6 pathogens-12-01245-f006:**
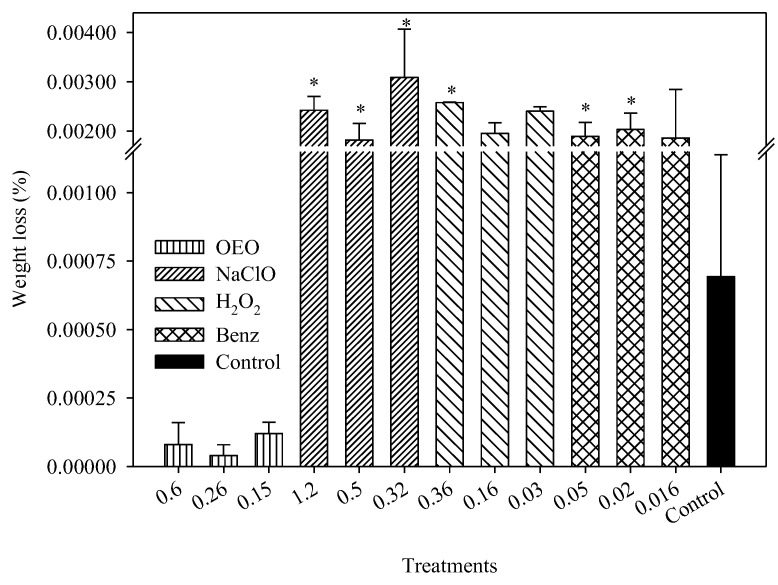
Weight loss of 304 stainless-steel exposed to different concentrations of OEO, NaClO, H_2_O_2_, and Benz after a 30-day regimen of a simulated sanitation protocol. * Significant differences (*p* < 0.05).

**Table 1 pathogens-12-01245-t001:** Minimum inhibitory concentration (MIC), minimum bactericidal concentration (MBC), minimum biofilm inhibitory concentration (MBIC), and minimum biofilm eradication concentration (MBEC) of OEO, NaClO, H_2_O_2_, and Benz against *S.* Typhimurium and *E. coli* O157:H7.

Bacteria		Compounds (mg mL^−1^)			
		OEO	NaClO	H_2_O_2_	Benz
***Salmonella* Typhimurium**	**MIC**	0.250	0.500	0.060	0.020
	**MBC**	0.300	0.600	0.080	0.025
	**MBIC**	0.150	0.450	0.030	0.022
	**MBEC**	0.300	1.200	0.160	0.050
** *Escherichia coli* ** **O157:H7**	**MIC**	0.220	0.400	0.140	0.020
	**MBC**	0.260	0.500	0.160	0.025
	**MBIC**	0.170	0.320	0.050	0.016
	**MBEC**	0.520	1.000	0.360	0.050

## Data Availability

All data related to this work are presented in the manuscript.

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
