# Peer review of "Oregano Essential Oil versus Conventional Disinfectants against Salmonella Typhimurium and Escherichia coli O157:H7 Biofilms and Damage to Stainless-Steel Surfaces"

_pathogens, 2023, doi:10.3390/pathogens12101245_

Round 1

Reviewer 1 Report

-          Page 3, section 2.2 and 2.4 need statistical values to support the data.

-          L 129-103, was the reduction statistically meaningful?

-          L229, how was the bacteria adjusted? Please describe in detail

-          L264, what is saline solution? Please describe composition and pH

Author Response

Dear reviewer, thanks for your observations for our manuscript, please find the responses below: 

Page 3, section 2.2 and 2.4 need statistical values to support the data. Las mics (2.2)

Thanks for your recommendation, ‘please find the improved text in lines 201-205

-          L 129-103, was the reduction statistically meaningful? Minimum inhibitory concentrations (MIC) and minimum bactericidal concentrations (MBC) are fundamental parameters in bacterial pharmacophysiology. MIC denotes the lowest concentration of an antimicrobial agent capable of retarding bacterial growth without necessarily causing the demise of all present bacteria. Conversely, MBC indicates the lowest concentration of an antimicrobial agent that can eliminate every bacterium within a given population. In this scenario, no bacterium survives exposure to the antimicrobial. The critical difference between MIC and MBC is their effect on bacteria, with MIC halting growth and MBC decisively eliminating bacterial populations. This comprehension is imperative in determining whether an antimicrobial is bacteriostatic (arrests bacterial growth) or bactericidal (lethal to bacteria). Understanding both MIC and MBC is crucial for optimizing antibiotic or antimicrobial therapy, preventing the emergence of antibiotic resistance, and selecting the most appropriate antimicrobial based on the nature of the infection and the patient's condition.

-          L229, how was the bacteria adjusted? Thanks for the observation, find the corrected text in the lines 92-95

-          L264, what is saline solution? Please describe composition and

Thanks for the observation, find the corrected text in the lines 129-130

Reviewer 2 Report

The manuscript entitled “Oregano essential oil versus conventional disinfectants against Salmonella Typhimurium and Escherichia coli O157:H7 biofilms and damage of stainless-steel surfaces” is a good work as disinfection of stainless-steel surfaces in the food industry is a challenging task due to damage to SS surface. However, the following points should be addressed,

Ø  Abstract- Before discussing about damage to stainless steel (line18-19), discuss about the antibacterial effect of the oil.

Ø  Keywords: Include biofilm and oregano oil

Ø  Lines 49-57: Why authors have described only about Artemisia oil, which is not directly related to the manuscript. Instead, in the introduction, give a short note on use of essential oils to combat bacterial biofilms in food safety establishments.

Ø  Give few lines about the importance of S. Typhimurium and E. coli O157:H7 in the introduction. Discuss their food safety and public health importance.

Ø  I think materials and methods should follow results and discussion.

Ø  In materials and methods, give more details about the oil used. If commercial preparation, furnish necessary details like the purity, solvent used to dilute? etc.

Ø  Line 228: replace the word “reactivated” as “revived”.

Ø  Why these strains have been chosen- Salmonella Typhimurium ATCC 14028 and Escherichia coli O157:H7 ATCC 43890. Are they strong biofilm formers, Explain.

Ø  Replace “non-brushing control” with some other suitable term throughout the manuscript

Ø  The subsections of the results section are highly confusing. Align with the main objectives of the study. Write the significant results (Microbial log reduction in MIC and MBC, and damage to SS surface)

Ø  Discussion: Rewrite. Highlight salient findings of the study and discuss previous studies on essential oil on biofilm and SS surface and compare

Ø  Conclusion: Rewrite.

Moderate editing of English language required

Author Response

Dear reviewer, thanks for your recommendations for our manuscript, please find the responser below:

Abstract- Before discussing about damage to stainless steel (line18-19), discuss about the antibacterial effect of the oil.

Thanks for the observation, find the corrected text in the lines 40-45, 68-70

Keywords: Include biofilm and oregano oil

Thanks for your review, please find the improved text in lines 28-29.

Lines 49-57: Why authors have described only about Artemisia oil, which is not directly related to the manuscript. Instead, in the introduction, give a short note on use of essential oils to combat bacterial biofilms in food safety establishments.

Thanks for the observation, find the corrected text in the lines 68-70.

Give few lines about the importance of S. Typhimurium and E. coli O157:H7 in the introduction. Discuss their food safety and public health importance.

Thanks for our recommendation, please find the improved text in lines 40-45.

I think materials and methods should follow results and discussion.

Thanks for your comment, the sequence of the manuscript was improved.

In materials and methods, give more details about the oil used. If commercial preparation, furnish necessary details like the purity, solvent used to dilute? etc.

Thanks for the observation, find the corrected text in the lines 84-86

 Line 228: replace the word “reactivated” as “revived”.

Thanks for your recommendation, please find the improved text in Line 92.

Why these strains have been chosen- Salmonella Typhimurium ATCC 14028 and Escherichia coli O157:H7 ATCC 43890. Are they strong biofilm formers, Explain.

Thanks for your recommendation, please find the improved text in Lines 40-46.

Replace “non-brushing control” with some other suitable term throughout the manuscript

Thanks for your observation, find the improved text in line 235 and figure 3.

The subsections of the results section are highly confusing. Align with the main objectives of the study. Write the significant results (Microbial log reduction in MIC and MBC, and damage to SS surface)

Thanks for your recommendation, please find the improved text in Lines 201-205.

Discussion: Rewrite. Highlight salient findings of the study and discuss previous studies on essential oil on biofilm and SS surface and compare

Thanks for your observation, please find the improved text in lines 280-293.

 Conclusion: Rewrite.

Thanks for your recommendation, please find the improved text in lines 340-346.